# Subcutaneous Myoepithelioma in the Extremity: A Potential Pitfall in the Differential Diagnosis of Subcutaneous Tumors

**DOI:** 10.3390/medicina59040667

**Published:** 2023-03-28

**Authors:** Minsun Koo, Young Chan Wi, Jimin Kim, Sheen-Woo Lee

**Affiliations:** 1Department of Radiology, Eunpyeong St. Mary’s Hospital, College of Medicine, The Catholic University of Korea, 1021, Tongil-ro, Eunpyeong-gu, Seoul 03312, Republic of Korea; 2Department of Pathology, Eunpyeong St. Mary’s Hospital, College of Medicine, The Catholic University of Korea, 1021, Tongil-ro, Eunpyeong-gu, Seoul 03312, Republic of Korea

**Keywords:** myoepithelioma, subcutaneous, soft tissue tumor, shoulder, ultrasonography, magnetic resonance image

## Abstract

We present a rare case of myoepithelioma in the subcutaneous layer of the shoulder with ultrasonography (US) and magnetic resonance imaging (MRI). US showed a lobulated hyperechoic mass, leading to an impression of lipoma. MRI showed the mass with low signal intensity on T1-weighted images (T1WI), high signal intensity on fat-suppressed T2-weighted images (T2WI), intermediate signal intensity on T2WI, and intense enhancement with adjacent fascial thickening. Imaging findings of soft tissue myoepithelioma have not been established. We report its US and MRI features mimicking features from a lipomatous tumor to infiltrative malignancy. Although soft tissue myoepithelioma has nonspecific image findings to confirm its diagnosis, some findings may help to make the differential diagnosis. Preoperative pathologic confirmation is recommended in a soft tissue neoplasm.

## 1. Introduction

Myoepithelial cell tumors are unusual neoplasms composed of myoepithelial cells. They most commonly occur in salivary glands and are thought to be due to the proliferation of myoepithelial cells between the epithelium and basement membrane. Soft tissue myoepithelioma is a rare entity that primarily presents in the upper and lower limbs [1,2]. To the best of our knowledge, none of the existing reports have included matching the ultrasonography (US) and magnetic resonance image (MRI) appearance. Here, we report the appearance of shoulder myoepithelioma on US and MRI, along with a review of the literature.

## 2. Case Report

A 69-year-old male patient presented with a slowly growing, painless mass on his right posterior shoulder noted four months ago. He had a history of total thyroidectomy with central lymph node dissection for papillary thyroid cancer, four years ago. Initial physical examination revealed an about 3 cm sized hard movable mass in the superficial layer of his right posterior shoulder. US presented a multi-lobular hyperechoic mass measuring about 2.8 cm, with internal echogenic striations in the subcutaneous layer of the right posterior shoulder. However, vascularity was not increased (Figure 1). The initial differential diagnosis from US included benign lipoma or well-differentiated liposarcoma. Due to the atypical ultrasonographic features, such as indistinct margination and echogenicity, a subsequent MRI was performed. The MRI showed an about 2.8 × 1.7 cm sized distinct mass with low signal intensity on T1-weighted images (T1WI) and mixed intermediate to high signal intensity on fat-suppressed and routine T2-weighted images (T2WI). The mass showed diffuse enhancement with thin septa and tail-like superficial fascial thickening on gadolinium-enhanced T1WI, suggesting involvement of adjacent fascia. There was no remarkable involvement of adjacent bones or muscles (Figure 2). Under the impression of locally infiltrative fibroblastic lesions such as nodular fasciitis, densely cellular superficial lymphoma, or myxoid liposarcoma, due to the sonographic appearance of fat, ultrasound-guided core needle biopsy followed by surgery was performed. On excision, the mass was well-encapsulated, mobile, and rubbery. The mass was removed with clear resection margins. Histologic examination revealed a neoplasm with a mixture of dense tumor cell proliferation, clear cell components, and collagenous septations (Figure 3). It was positive for actin, S-100, cytokeratin-AE1/AE3, and focally positive for epithelial membrane antigen (EMA), but negative for desmin, CD34, MDM2, p63, and HNB-45. The final diagnosis was myoepithelioma. Recent computed tomography (CT) follow-up, 9 months after the diagnosis, revealed no evidence of local recurrence.

## 3. Discussion

This is the first matched US and MRI report of myoepithelioma in the upper extremity. Myoepithelioma of soft tissue was first reported in 1997 [3]. Since then, it has been increasingly reported over the past decade. Clinically, soft tissue myoepithelioma remains unchanged as a palpable mass for several years with a peak incidence in the 3rd and 5th decades without sex predilection [4]. It occurs most frequently in the upper and lower limbs, followed by the trunk, and it may be localized as a superficial or deep mass resembling other similar lesions. It is usually superficial and localized in the subcutaneous layer [5,6]. The median size of tumor varies from 1 to 7 cm [1]. The World Health Organization (WHO) classified myoepithelioma as a tumor of uncertain differentiation, with myoepithelial carcinoma and mixed tumors [7]. The long-term prognosis of myoepithelial carcinomas is poor and local recurrence is frequent if the resection margin is incomplete [1].

Myoepithelioma is a tumor presenting a wide spectrum of epithelioid, spindled, histiocytoid, or plasmacytoid cells with myxoid or hyalinized stroma. Most reports show positivity for cytokeratin and variable positivity for epithelial membrane antigen (EMA). It is also positive for the S-100 protein in most cases. Myogenic marker expression, such as calponin, is positive in most cases, while smooth muscle actin (SMA) is positive in roughly half of cases [8]. Positive immunostaining for EMA with negative immunostaining for desmin assists in distinguishing spindle cell myoepithelioma from other smooth muscle tumors [9]. Although there have been reports of ESWR1 translocation in myoepithelioma or carcinoma [10], no genetic studies have been conducted in this case. Distinguishing myoepithelioma from myoepithelial carcinoma is very difficult because the immunohistochemical features of these two entities are similar. Differential points of myoepithelial carcinoma include positivity for p63 and a high degree of nuclear atypia [11]. Tumors with low-grade cytologic atypia are classified as myoepithelioma, while tumors with high-grade cytologic atypia are classified as myoepithelial carcinoma [12].

Soft tissue masses in the extremities, including myoepithelioma, are composed of a broad spectrum of benign and malignant lesions and they may mimic each other. Soft tissue masses can be distinguished according to their location in skin layers, including epidermis, dermis, subcutaneous and deeper layers, histological composition, and related abnormalities [13]. When we encounter a soft tissue mass, US is frequently used as a first-line imaging modality due to its easy availability and safety against radiation [14]. Certain superficial lesions, such as fatty tumors or benign cystic lesions such as epidermal inclusion cysts and ganglion cysts, have distinguishing characteristics that allow for a reliable diagnosis [15]. Lipoma, the most common soft tissue tumor, typically appears as an oval isoechoic to hyperechoic compressible mass with fibrous septa and no or minimal vascularity [15]. However, the accuracy of sonographic diagnosis is low, varying from 49% to 64%, due to the variable appearance [16]. Multiple studies’ findings have demonstrated that the ultrasound appearance of lipoma varies, ranging from anechoic to hyperechoic and well-defined to ill-defined. The echogenicity of a lipoma varies depending on its internal cellularity and increases as the number of fat–water surfaces increases [17]. A lipoma with atypical or any suspicious features should be further evaluated with MRI, if possible. Concerning clinical characteristics include rapid growth and pain in elderly patients. The presence of discernible internal vascularity, heterogeneous appearance, ill-defined or lobular margins, and size larger than 3 cm should raise a red flag [14,18]. Furthermore, there is significant overlap in US features among various superficial neoplasms, making differentiation between benign and malignant neoplasms difficult. Most neoplasms appear as a predominantly hypoechoic, solid or partially solid mass, with internal vascularity and variable heterogeneity. Hemorrhage and necrosis can make heterogeneous echogenicity and a cystic portion. Myxoid tumor may appear as a hypoechoic mass mimicking a fluid collection [15]. Furthermore, US is operator dependent; therefore, a poor beam adjustment and/or a deeply located lesion may result in a suboptimal examination [14,19]. Since ultrasound has such limitations in soft tissue masses, any atypical feature necessitates an MRI with contrast enhancement, followed by biopsy and surgery [18].

MRI is widely regarded as the best imaging method for evaluating soft tissue masses due to its high tissue contrast, ability to demonstrate relationships between the mass and adjacent structures, and to indicate specific components using multiple sequences [14]. If MRI shows that the lesion contains fat, the diagnosis can be narrowed down to adipocytic tumors and benign tumors containing fat [20]. Liposarcoma, the second most common soft tissue sarcoma, is classified histologically into five major types: well-differentiated, dedifferentiated, myxoid, pleomorphic, and spindle cell liposarcoma. Well-differentiated liposarcomas may sometimes mimic a benign lipoma on US, but they are usually in deeper locations and more hyperechoic with more detectable vascularity [15]. Additional features that suggest liposarcoma rather than lipoma include larger size, older patient age, presence of thick septa, decreased fat composition, presence of nodular or globular areas, and presence-associated masses [21]. Liposarcoma usually appears as a polylobulated tumor with small satellite nodules on MRI and the fatty signal varies depending on the degree of differentiation. Dedifferentiated, myxoid, and pleomorphic types may not show a macroscopic fat component on imaging and myxoid liposarcoma shows a T2 hyperintense pseudocystic signal with intense enhancement [20]. If there had been a macroscopic fat component, we would have diagnosed our case as a lipomatous tumor similar to US. In the absence of any macroscopic fat component, MRI can be useful in differentiating the melanocytic, myxoid, or fibrous component. Clear cell sarcoma, which is a melanocytic sarcoma, shows a T1 hyperintense melanin signal in about half of cases [20]. Deep soft tissue tumors containing a myxoid signal include intramuscular myxoma, myxoid chondrosarcoma, myxofibrosarcoma, myxoid liposarcoma, or intramuscular neurogenic tumor. They show high signal intensity on T2WI and enhancement after injection of contrast media [20]. Soft tissue tumors with a fibrous component include nodular fasciitis, fibrosarcoma, desmoid tumor, and inflammatory myofibroblastic tumors. Nodular fasciitis, the most common fibrous soft tissue tumor, shows intermediate signal intensity on T1WI and heterogeneous high signal intensity on T2WI with diffuse or peripheral enhancement [20]. MRI imaging features that suggest malignant soft tissue tumors include maximum diameter larger than 50.5 mm, unsmooth margin, fascial edema, skin thickening, skin contact, hemorrhage, necrosis, lobulation, and peritumoral edema [22,23]. Diffusion-weighted MR imaging may be useful in differentiating abscesses from necrotic tumors and benign from malignant tumors [20]. The decision on how to evaluate and manage the soft tissue mass varies depending on patient and tumor factors. Therefore, communication between clinicians and radiologists is important and a multidisciplinary approach is needed, especially in the management of soft tissue sarcoma [20].

Although several cases have been reported so far (Table 1), the radiologic features of soft tissue myoepithelioma have not been established yet. A clinical report of recurrent myoepithelial carcinoma of the leg included an ultrasound image, which suggested that the lesion was in the subcutaneous layer with a lobulated heterogeneous hyperechoic appearance [24], as in our case. Hashimoto, et al. reported a case of soft tissue myoepithelioma at the shoulder, between the supraspinatus and trapezius, with low signal intensity on T1WI, high signal intensity on T2WI, and heterogeneous enhancement. It also showed high concentrations of ^18^F-fluorodeoxyglucose [25]. In another case report, myoepithelial carcinoma in foot plantar soft tissue showed a T1 isointense signal and a T2/short tau inversion recovery heterogeneous hyperintense signal with intense enhancement [26]. However, myoepithelioma in the soft palate appeared as a well-defined, round, and partially lobulated mass with slight marginal enhancement on CT [27]. It showed a heterogeneous isointense signal compared to the pharyngeal muscle on T1WI, a heterogeneous hyperintense signal on T2WI, and a heterogeneous intense enhancement after administration of contrast media, similar to our case. These findings indicate soft tissue myoepithelioma and soft palate myoepithelioma share similar image findings. In a previous study comparing CT findings of two different soft palate myoepitheliomas [28], the tumor composed of plasmacytoid cells with a rich myxoid stroma showed faint enhancement while the cellular tumor with a fibrous stroma showed intense enhancement. These findings suggest the enhancement pattern of myoepithelioma is affected by its histological components. A microscopic examination of our case showed dense tumor cell proliferation with only focal clear cell changes, and these histological components may have contributed to the intense enhancement of the tumor after the administration of contrast material in MRI.

Myoepithelioma is difficult to diagnose with a single imaging modality due to nonspecific radiological findings and overlapping features from benign to malignant tumors. Thus, when we encounter a lobulated hyperechoic mass on US which shows T1 low and T2 intermediate to high signal intensity on MRI with diffuse enhancement and infiltrative features, we should consider including myoepithelioma as well as its malignant counterpart in the differential diagnosis. Therefore, acquiring a pathologic specimen is strongly recommended [29].

## 4. Conclusions

This case report presents a subcutaneously located soft tissue myoepithelioma of the shoulder with US, MRI, and histopathological findings. When approaching an echogenic lobulated soft tissue mass, we should be familiar with its differential diagnosis including myoepithelioma.

## Figures and Tables

**Figure 1 medicina-59-00667-f001:**
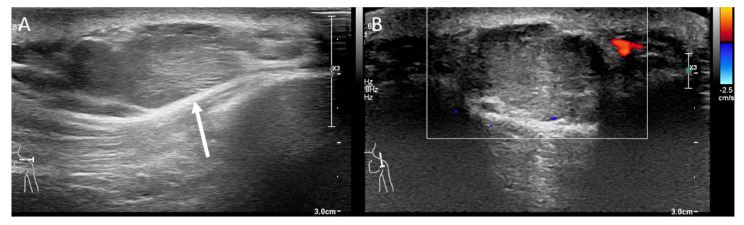
Transverse ultrasound scan shows lobulated hyperechoic mass (arrow) with internal striations in the subcutaneous layer of the posterior shoulder (**A**). Longitudinal scan with Doppler shows the mass without significant internal vascularity (**B**).

**Figure 2 medicina-59-00667-f002:**
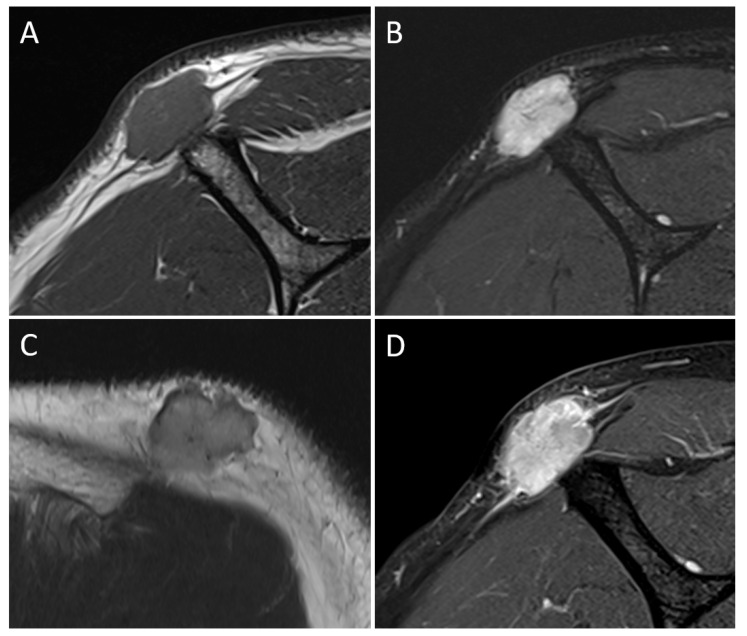
The sagittal view of the shoulder shows the lobulated mass with low signal intensity on T1-weighted images (**A**), and high signal intensity on fat-suppressed T2-weighted images (**B**). On T2-weighted images, the lesion is intermediate signal intensity but higher than the adjacent muscle (**C**). The lesion shows intense enhancement with adjacent fascial thickening (**D**).

**Figure 3 medicina-59-00667-f003:**
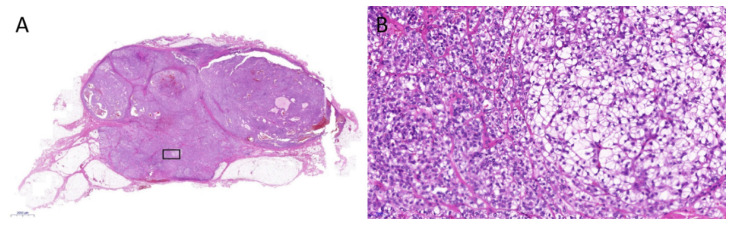
A microscopic view (hematoxylin–eosin stain, ×0.6 in (**A**), ×20 in (**B**), (**B**) is magnified view of the black box in (**A**) of the excised specimen shows dense tumor cell proliferation in collagenous septations and focal clear cell changes.

**Table 1 medicina-59-00667-t001:** Imaging features of soft tissue myoepithelial tumors.

Author, Year	Age (Years) /Sex	Location	Size (cm)	Pathologic Diagnosis	US Findings	MRI Findings
Rastrelli et al., 2013 [24]	61/M	Leg, subcutaneous layer		Myoepithelial carcinoma	Lobulated heterogeneous hyperechoic mass	
Hashimoto et al., 2020 [25]	72/W	Shoulder, intramuscular	8.3 × 6.5	Myoepithelioma		Mass with low signal intensity on T1WI, high signal intensity on T2WI, and heterogeneous enhancement
Trevino et al., 2020 [26]	12/M	Plantar foot, intramuscular	5 × 2	Myoepithelial carcinoma		Mass with intermediate signal intensity on T1WI, heterogeneous high signal intensity on T2WI/STIR, and intense enhancement
Current case	69/M	Shoulder, subcutaneous layer	2.8 × 1.7	Myoepithelioma	Lobulated hyperechoic mass with internal striations and no internal vascularity	Mass with low signal intensity on T1WI, high signal intensity on fat-suppressed T2WI, intermediate signal intensity on T2WI, and intense enhancement with adjacent fascial thickening

M; male, F; female, T1WI; T1-weighted images, T2WI; T2-weighted images, STIR; short tau inversion recovery.

## Data Availability

All data relevant to the study are reported in the manuscript.

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
