# Peer review of "Subcutaneous Myoepithelioma in the Extremity: A Potential Pitfall in the Differential Diagnosis of Subcutaneous Tumors"

_medicina, 2023, doi:10.3390/medicina59040667_

Round 1
Reviewer 1 Report
- I do not agree with the first statement in the introduction that myoepithelial cell tumors are relatively newly defined.
- The second statement refers to the origin of myoepithelial neoplasms in the salivary gland or breast. Tumors that arise in soft tissue lack any known normal cellular counterpart.
- What was the differential diagnosis based on the ultrasound?
- The discussion claims that reliable benign masses that can be diagnosed with confidence include lipoma. Why was an MRI done in this case if the ultrasound suggested lipoma?
- What was the core biopsy called? Was it different than the excision?
- The histologic description notes a “biphasic” tumor. What does that mean? Was there a ductal component?
- A subset of soft tissue myoepitheliomas are known to harbor EWSR1/FUS rearrangements or homozygous deletion of the SMARCB1. Please mention this in the text. Are the genetics known for this case?
- In soft tissue, what are the criteria for classifying lesions as malignant? Please indicate whether those features were present in this case.
- Is there a difference between ‘malignant soft tissue myoepithelioma’ and myoepithelial carcinoma? Please clarify the use of these terms in the text.
- What was the status of the resection margins?
- The discussion seems to lack focus. Please revise to focus on differentials relevant to this case based on the specific imaging findings.
Reviewer 2 Report
Hi dear Authors;
It is a well written and discussed article with radiological and histopathological correlations. It can contribute to the literature. Congratulations. However, the following minor suggestions must be made. In particular, number 3 is essential for to be published of the article and to comply with ethical rights and rules. please pay attention to this! Kind regards…
1-In the terminology where tumors are defined, myoepitheliomas are called cutanous and soft tissue myoepitheliomas. Myoepithelioma is uncommon in the subcutaneous tissue. Clinically, the neoplasm is nonspecific (Reference: Kadlub N, Galiani E, Fraitag S, Boudjema S, Vazquez MP, Coulomb A, Picard A. Soft tissue myoepithelioma of the scalp in a 11-year-old girl: a challenging diagnosis. Pediatr Dermatol. 2012 May-Jun;29(3):345-8. doi: 10.1111/j.1525-1470.2011.01428.x. Epub 2011 May 25. PMID: 21615483.) Based on this literature data, the present article is valuable for the literature.Therefore, the title of the Article can be changed as follows or a more appropriate one. For exemple;
Subcutanous (or subcutanous superficial soft tissue) Myoepithelioma of the shoulder: A Potential Pit[1]fall in the Differential Diagnosis, case report
2- Keywords should be renewed in accordance with the new title. For exemple;
Keywords: myoepithelioma; subcutanous; soft tissue tumor; Shoulder; ultrasonography; magnetic resonance image
3-Ok, It is a radiologic article but The pathologist is very important for the final diagnosis. The name of the pathologist who made the diagnosis should be included in the current article due to academic ethical rules. Moreover, histopathological photographs of the case are also included in the article, and it is indispensable to include the pathologist's name.
If the pathologist's name who was reported this case is not wanted to be added; In addition to the histopathological photographs, the detailed histomorphological description of the case, which is included in the case report section (Histologic examination revealed a biphasic tumor with clear cell change (Figure 3). It was positive for actin, S-100, cytokeratin-AE1/AE3, and focally positive for epithelial membrane antigen (EMA), but negative for desmin, CD34, MDM2, p63, and HNB-45. The final diagnosis was myoepithelioma) and taken from the pathology report, should be removed from the article.
4-Myoepitheliomas can be skin or soft tissue localized. Soft tissue localized ones can be deep or superficial localized. Thats why the sentence in line 5. in dıscussion part (It occurs most frequently in the upper and lower limbs, followed by trunk, and it may resemble other superficial lesions. It is usually su[1]perficial and localized in the subcutaneous layer.) can be changed, like this: It occurs most frequently in the upper and lower limbs, followed by trunk, and it may localised as superficial or deep mass resemble other similar lesions. It is usually superficial and localized in the subcutaneous layer. (some references about it: Plaza JA, Brenn T, Chung C, Salim S, Linos KD, Jour G, Duran Rincon J, Wick M, Sangueza M, Gru AA. Histomorphological and immunophenotypical spectrum of cutaneous myoepitheliomas: A series of 35 cases. J Cutan Pathol. 2021 Jul;48(7):847-855. And Kadlub N, Galiani E, Fraitag S, Boudjema S, Vazquez MP, Coulomb A, Picard A. Soft tissue myoepithelioma of the scalp in a 11-year-old girl: a challenging diagnosis. Pediatr Dermatol. 2012 May-Jun;29(3):345-8. )
5- In terms of manuscript integrity, some words in the conclusion section can be edited. Like this: This case report presents superficial/subcutanous located soft tissue myoepithelioma of the shoulder with US ,MRI and histopathological (‘Histopathological’ words should be added here ıf pathologist name is addeed the article) findings. When approaching an echogenic lobulated soft tissue mass, we should be familiar with its differential diagnosis including myoepithelioma.
Round 2
Reviewer 2 Report
Hi dear Authors;
you have meticulously made all the suggestions. thank you. In its current form, it is a well-written article and may contribute to the literature. Sincerelly